# Environmental and Management Factors Affecting the Time Budgets of Free-Ranging Iberian Pigs Reared in Spain

**DOI:** 10.3390/ani10050798

**Published:** 2020-05-05

**Authors:** Míriam Martínez-Macipe, Eva Mainau, Xavier Manteca, Antoni Dalmau

**Affiliations:** 1IRTA Veïnat de Sies S/N, 17121 Monells, Spain; sundarimmm@gmail.com; 2UAB, Veterinary School, Campus Universitat Autònoma de Barcelona, 08193 Cerdanyola del Vallès, Spain; eva.mainau@uab.cat (E.M.); xavier.manteca@uab.cat (X.M.)

**Keywords:** free-range pigs, Iberian pigs, pig behaviour, pig welfare, time budgets

## Abstract

**Simple Summary:**

Understanding the natural behaviour of pigs in free-range conditions facilitates the interpretation of their behaviour in intensive conditions. The present study aims to study behavioural indicators of activity in the domestic pig, reared free-range and under two feeding regimes, with some help from humans with concentrates and without help, just eating natural resources. Results confirmed that exploratory behaviour was an important behaviour for pigs, but the same animals that dedicated 50% of their time to this activity when they were not fed by humans reduced this activity to 17.8% when they were fed with concentrates. In addition, few social contacts between animals were seen in extensive conditions, with a higher incidence of negative rather than positive social behaviour. It was concluded that: (1) the need for exploring the surroundings in natural environments is of less importance for pigs when they are fed by humans, and (2) bathing areas in outdoor conditions are important for pigs in the event of warm conditions. Overall, it is concluded that natural behaviour of pigs in free-range conditions, such as a reduced foraging behaviour when pigs are fed with concentrates, should be considered when interpreting behavioural needs in intensive conditions.

**Abstract:**

Understanding the natural behaviour of pigs in free-range conditions facilitates interpretation of their behaviour in intensive conditions. Studying six different farms over two years at different seasons, with climatic and management variations, allowed for a general description of Iberian pig behaviour and which factors have an influence on it. The main activity found was resting (56.5% of the time observed), followed by exploratory behaviour. However, this exploratory behaviour was higher when animals were fed only with natural resources than when fed with concentrates (50% versus 17.8%, respectively). In addition, pigs used bathing areas in summer that were not visited in winter. Negative social behaviour was seen more frequently than positive social behaviour, accounting, in total, for 1% of the total activity of animals. Pigs situated at the centre of the groups tended to remain more relaxed, while the peripheral animals remained more alert and vigilant. Our results indicate that foraging behaviour accounts for a significant proportion of pigs’ active time, but this proportion is much more reduced when pigs are fed concentrates. Therefore, behavioural needs in pigs reared in intensive conditions should consider that exploratory behaviour is reduced when pigs are fed with concentrates.

## 1. Introduction

The Iberian pig is an autochthonous breed, bred in the southwest of the Iberian Peninsula (Spain and Portugal) [1] for high-quality meat products. Currently, the meat is considered “Iberian” when it comes from a pig with a minimum of 50% Iberian genetics, where the mother must be 100% Iberian (RD 4/2014). In the traditional production system, Iberian piglets are weaned at two months old, and then they are usually mixed in large pastures. In these areas, they are given concentrate at the same time as getting natural resources for several months, until reaching around 90–115 kg of body weight [2]. However, during this initial fattening phase, some farms of Iberian pigs prefer to keep their pigs inside and feed them only with concentrate. The late fattening phase is performed, in all cases, free-range, with the dehesa agrosystem, a clear forest of evergreen oaks, *Quercus rotundifolia* [3]. In winter, when the acorns from the oaks (*Quercus ilex*) and cork oaks (*Quercus suber*) fall, pigs eat only the acorns and other natural products like tubers, fungi or roots from the dehesa pastures. This late fattening phase at the dehesa is called “montanera” and the pigs must gain their last 50–60 kg based only on that type of food source [4]. Iberian pigs are usually slaughtered at 14–24 months of age, with a body weight of 150–160 kg (RD 4/2014).

Years after the Farm Animal Welfare Council included the concept of “expression of normal behaviour” as one of the five freedoms [5], the Welfare Quality protocol established behaviour as the fourth principle to assess welfare at the farm level [6]. It has been applied in order to evaluate the welfare state of pigs [7] and other species, such as broilers [8], cows [9] and even buffalos [10]. As behaviour is regarded as a welfare indicator, it is important to know how Iberian pigs behave in their traditional environment to be able to detect welfare problems in the future. The natural environment provided to Iberian pigs might be regarded per se as a cognitive enrichment, which should enhance the animals’ welfare [11]. The study of behaviour is useful to discover general habits and activities in these animals and this is essential to evaluate the welfare of the animals, which is crucial for productive efficiency [12]. For instance, pigs are strongly motivated in performing exploratory behaviour which, in nature, is necessary to search for food or to gather general information on their surroundings [13]. However, there is a lack of research in (semi-) natural conditions to be able to confirm if a wallowing behaviour could be an important behaviour for pig welfare [14]. Environmental and management factors, such as dietary supplementation or castration, may change the expression of some behaviour patterns of Iberian pigs reared outdoors, such as looking for food or mating behaviour.

Iberian pigs have two different types of feeding schedules, depending on the stage of the year: first, they rely on the farmers in order to get food, and then they need to search by themselves in order to find food resources from the ground. This change in feeding patterns would probably affect their activity budget, as feeding necessity probably increases foraging activity during the montanera period [7]. Meteorological conditions might affect the behaviour of Iberian pigs when they are reared free-range, as would happen with other breeds in semi-extensive systems [15].

Pigs are social animals and usually live in groups [16], performing a wide repertoire of sexual, feeding and social behaviours [17]. In many species living in groups, the position they have within the group determines their role and the behaviour they will be exhibiting. For example, in some species of ungulates, the dominants are located in the centre and leave the subordinates at the periphery, more exposed to danger and spending more time alert in order to detect the presence of a predator [18]. In the chamois (*Rupicapra pyrenaica*), females prefer secure areas, especially when their offspring are more vulnerable, producing a sexual segregation [19]. In the case of Iberian pigs reared for consumption, all animals in the group have the same age and no offspring are present. However, in wild boars (*Sus scrofa*), synchronised vigilance is related to group size and risk factors [20].

In the traditional production system for the Iberian pig, both males and females are castrated. In the case of males, to avoid boar taint after slaughter, and in the case of females, to avoid wild boars being attracted to the enclosure by entire females in heat. However, spaying is regulated by Spanish national law (RD 1221/2009), and the castration of males will be limited in the EU. Consequently, alternatives to this surgery such as vaccination against gonadotropin releasing factor (GnRF) (immunocastration) are being studied. Immunocastration has long been used in intensively bred white pigs [21] and may also be a good option for free-ranging Iberian pigs [22]. Immunocastration could suppress oestrus and sexual behaviour [23,24]. Oestrus is characterised by a marked increase in social activities such as snouts contacts, ano-genital sniffing and flank nosing and mounting [25]. Immunocastrated pigs show less non-violent social and aggressive behaviours than entire male pigs [24].

In the last few decades, outdoor systems for Iberian pigs have been changed, using a more intensive system. Some of these changes mean the shortening of the production system, fattening with mixed feed and the impossibility of pigs to perform some natural behaviours [3]. All of these changes may involve a detrimental effect on animal welfare. One of the main issues is the lack of knowledge of the needs of pigs outdoors, that should be reproduced indoors. 

While there are many research articles related to pig behaviour, only 0.46% of them were done using Iberian pigs (ScienceDirect search, 2018), and most of them are focused on productive improvement. Consequently, there is little information available about what Iberian pigs do in the dehesa and which factors affect their behaviour. Which behaviours seem more important for pigs? Do they often interact socially? Does the weather change their daily activities? Do they use a water bathing area? Does the position within the group affect their behaviour? Does their sexual condition affect their behaviour? Do they change their activity budget when they receive dietary supplementation by humans, in comparison to the montanera period when they receive no supplementation? 

The aim of this study is to investigate the factors influencing the activity of Iberian pigs reared outdoors at specific moments of the day. The first step of the study was to describe the activity of Iberian pigs reared outdoors. The second step was to study the effects of environmental (weather and season), management (dietary supplementation and castration) and intrinsic factors (position of the animal within a group and sex) on the activity of Iberian pigs. 

Results should be taken under consideration for better knowledge of Iberian pigs and for better adaptation to other rearing conditions.

## 2. Materials and Methods

All data were collected at six different farms, dedicated to Iberian pig production in Extremadura, Spain. The study was performed during two consecutive production cycles (from spring, when they were weaned, until slaughter the following winter, after the montanera period), from 2012 until 2014. Consequently, each pig was studied for around 12 months. All pigs in the study were Iberian breed pigs with differing percentage levels of Iberian breed, depending on the farm. The quantity of animals studied varied with the farm and production cycle, as the whole group was included in the observations (Table 1). At all farms, there were both males and females neutered by surgery (50 males and 30 females and 18 males in Farm 1 the first and second year, respectively; 110 males and 90 females in Farm 2; 60 males and 67 females and 45 males and 42 females in Farm 3 the first and second year, respectively; 68 males and 72 females and 108 males and 92 females in Farm 5 the first and second year, respectively; 62 males and 56 females in Farm 6). In addition, at Farm 4, there were also entire and immunocastrated animals (20 castrated males, 23 entire males, 19 immunocastrated males, and 11 castrated females, 46 entire females and 26 immunocastrated females). Some of them were transferred to other farms. Consequently, for the montanera season, 12 castrated and 3 immunocastrated males and 6 castrated, 4 entire and 18 immunocastrated females remained. At Farm 1, there were 14 immunocastrated females in the first year and 22 in the second year. 

Before the montanera period and from when they were weaned, pigs were reared in a mixed-sex group in an open-air enclosure, with a refuge to sleep in or go to whenever they wanted, with a cement area to receive food, two access points for water and a basin to bathe in. The animals were never mixed with unknown animals after weaning. The enclosures occupied from 3 to 12 Ha and included oaks and other normal dehesa vegetation. At all farms, animals were fed pellets once a day, in the morning, except for Farm 6, which also incorporated tomato pulp in the afternoons.

When the acorns started to fall in early winter, the montanera period started, and pigs were led to a bigger enclosure, from 12 to 180 Ha, or their enclosure was opened to the field next to it; consequently, they had more space to range freely. At some farms, they opened new enclosures or changed animals from one field to another, depending on acorn availability. In these cases, they ate all acorns from one place first and then from the other, in that order. During this period, animals were not fed by farmers at all, they just ate the natural products from the dehesa environment. 

Each farm was visited once every one or two weeks, mostly from 7:30 h to 15:00 h. Eight percent of the observations were carried out from 07:30 h to 09:30 h, 56% from 09:30 h to 11:30 h, 27% from 11:30 h to 13:30 h, 2% from 13:30 h to 15:00 h and 7% from 17:00 h to 21:00 h, avoiding the times where animals were fed in the pre-montanera period. In total, visits occurred on 133 different days. To observe the animals, the same person travelled every day to the different farms, by van, and entered the animals’ enclosure on foot. If the animals perceived the presence of the observer (assessed by means of a change or cessation in the activity of different individuals in a group, with the animals raising their heads), the person waited for 10–15 min before starting to register the behaviours, giving them time to get used to his presence. When possible, the person observed from a distance and followed the animals, trying not to be noticed by using binoculars. An observation period consisted of two hours of observation, combining the techniques of scan and focal sampling [26]. The scan consisted of a picture taken every ten minutes and considering the activity of all of the animals in a group, which were registered individually in one of the pre-established behaviours shown in Table 2. In the best of cases, 12 consecutive scans could be achieved in two hours. However, some of them were lost due to the movement of the animals to areas with a worse view during the period of two hours. A total of 1439 scan samples (with all animals of the group assessed) were finally recorded in the study (310, 31, 468, 162, 290 and 177 for Farms 1, 2, 3, 4, 5 and 6, respectively). During the lapse of time of 10 min between two scans, two consecutive focal samplings of 5 min each were carried out. In this case, and using the same descriptors of Table 2, the observer focused on a continuous observation on one of the animals of the group. This animal was chosen according to its position in the group (central or peripheral; 50% of each type) and randomly according to sex or type of castration (none, conventional or immunocastration). Central was an animal surrounded by others, and peripheral was an animal at the margins of the group or the isolated animals. For 5 minutes, the behaviour of the animal was continuously narrated and recorded with a voice-recording device. Only those animals clearly visible, with a complete time of observation for the 5 min and clearly in a central or peripheral position were studied. In addition, in case of doubts about a possible repetition of the same animal due to the movements of the group, focal samplings were cancelled for this specific day. In consequence, although a maximum of 24 focal samplings were possible for an observation period (120 min for 5 min each), rarely was this number achieved. In fact, a total of 1247 focal samplings were finally recorded in the study (286, 32, 393, 131, 261 and 144 for Farms 1, 2, 3, 4, 5 and 6, respectively). 

At the beginning and at the end of the visit, the weather conditions (temperature, humidity and wind speed) were measured with an Amprobe TMA40-A (Glottertal, Germany) meteorological station. The temperature-humidity index (THI) was calculated for each day of observations according to the following formula:

THI = Temperature − [(0.31 − 0.31 × Humidity/100) × (Temperature − 14.4)] [28].

The means, minimum and maximum THI measures of each season and the mean wind speed are summarised in Table 3. As for the weather conditions, according to the sky and visibility registers, days were classified as sunny (84%), cloudy (7.2%), partly cloudy (4.3%), rainy (3.3%) and foggy (1.2%). According to wind speed, they were classified as non-windy (86.5%, from 0 to 5 km/h) or windy (13.5%, from 5 to 11 km/h, corresponding to a 1–2 value of the Beaufort scale).

For the observation of the animals, binoculars were used when necessary with a tripod. For keeping records, a Sony DSC-W350 (Barcelona, Spain) camera and an Olympus VN-712PC (Barcelona, Spain) voice recorder were used. 

### Statistical Analysis

Statistical analyses were carried out with the Statistical Analysis System (SAS software, SAS Institute Inc.; Cary, NC, USA). For the data coming from the scan samplings, where the activity of different animals were registered at the same time, what was considered for each scan is the percentage of animals performing a specific behaviour at a certain moment (% of animals performing exploratory walking, exploratory standing, resting, social positive, social negative, walk, run, bathing, gazing, drinking). For the data coming from the focal samplings, which were taken individually on the animals, the % of time dedicated by the animal to the different behaviour for each focal sampling was considered individually. In both cases, this was the data used for the statistical analysis. In the case of scan sampling, the unit was the group of animals observed, and in the case of focal sampling, the unit was the individual observed. Each behaviour was assessed separately, taking into account the effect of the montanera (yes or no), position in the group (central or peripheral), season (spring, summer, autumn or winter), weather (sunny, rainy, foggy, partly cloudy or cloudy), sex and presence of wind (yes or no) as fixed effects. The farm (1 to 6) was also included in the model as a fixed effect. General models, with a Poisson or negative binomial distribution, according to Cameron and Trivedi [29], were used. The least square maximum likelihood was used as a method of estimation. The least square means of fixed effects (LSMEANS) were used when the analysis of variance indicated differences. In all cases, significance was fixed at *p* < 0.05.

## 3. Results

### 3.1. Activity

In mean values (mean ± SE), animals were seen 56.5% ± 0.90% of the time resting, 20.9% ± 0.64% exploring standing, 7.6% ± 0.36% exploring walking, 5.8% ± 0.27% walking, 4.3% ± 0.22% bathing, 1.5% ± 0.12% gazing, 1.1% ± 0.14% running, 1.0% ± 0.07 % socialising (0.4% ± 0.03% positive, 0.6% ± 0.05% negative), 1.0% ± 0.07% drinking and 0.2% ± 0.02% performing other behaviours.

### 3.2. Effect of Being Fed or Not, the “Montanera Effect”

Exploratory behaviour, including walking and standing, had a clear “montanera” effect (Chi-square = 37.80; Degrees of Freedom (DF): 1; *p* < 0.0001 and Chi-square = 43.70; DF: 1; *p* < 0.0001). Bathing and drinking were also affected by the montanera (Chi-square = 44.90; DF: 1; *p* < 0.0001 and Chi-square = 6.55; DF = 1; *p*= 0.01, respectively). Exploring walking (17.9%) and exploring standing (32.1%) were higher during the montanera period than during the rest of the year (11.3% and 6.5%, respectively). Bathing was observed in 5.5% of the observations outside of the montanera period, and never during this period. Drinking behaviour was also seen less (0.2%) during the montanera, as compared to the rest of the year (1.2%).

### 3.3. Effect of Season

Both exploring walking and standing were affected by season (Chi-square = 20.84; DF = 3; *p* < 0.0001 and Chi-square = 40.06; DF = 3; *p* = 0.0002, respectively). Social negative, bathing and drinking were also affected (Chi-square = 15.36; DF = 3; *p* = 0.0009, Chi-square = 66.50; DF = 3; *p* < 0.0001 and Chi-square = 17.90; DF = 3; *p* = 0.0004, respectively).

In winter, the season of “montanera” had the highest percentages for exploring walking and exploring standing (Figure 1). The highest percentage for bathing was seen in summer. The season where there was more social negative behaviour was spring (1.1%), followed by summer (0.7%), winter (0.6%) and autumn (0.4%). Animals drank, in a higher percentage, during summer (1.5%), followed by spring (1.0%), autumn (0.7%) and, finally, winter (0.2%).

### 3.4. Effect of Weather

The weather affected the time that animals were resting (Chi-square = 21.23; DF = 4; *p* = 0.0003), showing social negative (Chi-square = 22.74; DF = 4; *p* = 0.0001), and social positive behaviour (Chi-square = 11.39; DF = 4; *p* = 0.02) and drinking (Chi-square = 47.63; DF = 4; *p* < 0.0001). They were seen resting 82.8% of the time during foggy days, 56.8% on sunny days, 49.6% on cloudy days, 70.6% on partly cloudy days and 38.7% on rainy days. 

Animals showed more social positive behaviour on cloudy days than on foggy days and more negative social behaviour on cloudy days than on sunny days (Figure 2). Finally, drinking was observed 1.2% of the time during sunny days, 0.4% on rainy days, 0.2% on cloudy and partly cloudy days, and it was not seen at all during foggy days (0%). 

Bathing was not affected by the sky’s situation, but rather by the presence of wind (Chi-square = 10.46; DF = 1; *p* = 0.00012): During windy days, animals were seen bathing only 0.7% of the time, while this percentage increased to 4.9% on non-windy days. In relation to THI, when ranged from 6 to 12, animals were seen bathing on only one occasion (0.01%), being 0.62% when ranging from 13 to 19, 4.41% when ranging from 20 to 26 and 12% when ranging from 27 to 33.

### 3.5. Position in the Group

The position of the animal in the group affected resting (Chi-square = 4.70; DF = 1; *p* = 0.03), explore walking (Chi-square = 14.94; DF = 1; *p* = 0.0001), walking (Chi-square = 21.43; DF = 1; *p* < 0.0001) and gazing (Chi-square = 15.78; DF = 1; *p* < 0.0001). Animals located at a central position displayed less resting behaviour, exploring walking, walking and gazing than did peripheral animals (Figure 3). 

### 3.6. Effect of the Sex

There were no significant differences for any of the behaviours when compared between males and females. 

## 4. Discussion

During the observation periods of the present study, Iberian pigs spent a mean of 56% of the time resting and 28.5% dedicated to exploratory behaviours, similar to what Horsted et al. [30] found in Denmark in cross-bred free-range pigs, where the animals spent 54.4% of the time resting, 19.3% rooting and almost 8% eating allocated feed. Even when, in the Danish study, temperatures were lower than the minimum temperatures registered in the present study, the similarities seem to be a good indicator of what the predominant activity budget is of free-range pigs in general. Guy et al. [31] found that the main activities when pigs were at outdoor paddocks were rooting and exploring. Hodgkinson et al. [32] and Rivero et al. [33] found that wild boars spent more or less the same amount of time resting as foraging (around 42%–43%) when groups of captive-bred animals were reared free-range only during daylight hours. In both cases, the animals may have been more motivated to forage than to lie down during the day, as they had to spend the nights in environments less enriched. However, another important point to take into account is whether the animals are fed or not. In the present study, animals were fed with concentrate for one part of the year, but they had to look for their own food the other part (montanera). During the montanera, exploratory behaviour (that, in the present study, could include foraging, licking, sniffing and even rooting because of the difficulties to discriminate among these behaviours from a distance without disturbing the animals) reached 50.0%, similar to the 54% described by Rodríguez-Estévez et al. [1]. Nevertheless, in the present study, this behaviour was previously found to be 17.8%, in the same animals, when they were fed with concentrates. Therefore, when defining activity budgets and behavioural needs in pigs, it should be considered whether these animals need to look for food or not. Another point to consider is the age of the animal, as in the present study, even inside the period with supplementation, it was observed that in spring, when the animals were younger, they explored more than in autumn or summer. 

In the present study, pigs were seen in the water 4.3% of the time. Taking into consideration that they did not do this at all in winter (when temperatures around 8 to 12 °C are normal), access to a bathing area should be considered an important resource for Iberian pigs during the warm periods (8.1% of the total activity budget in summer, when temperatures can reach values over 35 °C). In fact, below a THI of 20, animals seen bathing did not reach a mean value of 1%, but above 27, the mean value was 12%. Pigs usually wallow in the mud in order to deworm, protect themselves from the sun and regulate body temperature [14]. When access to bathing areas is ensured, they show a clear preference for these areas to be refreshed. However, according to the observations of the present study, pigs did not use water bathing areas during the montanera. In addition, due to the climatic conditions, the bathing areas for pigs presented a worse state in summer (the driest and warmest season) than during the montanera season. Therefore, it is important to highlight how essential it is for pigs to have these bathing areas in excellent conditions in summer. Consequently, in the same way that efficient facilities try to reduce thermal stress (i.e., ventilation, refreshment systems) in indoor systems, the provision of bathing areas should be provided for pigs in outdoor conditions during the warmest seasons. Stolba and Wood-Gush [34] had already stated that housing conditions should be tailored to the behavioural needs of pigs, as, if applied wisely, it would still leave room for economic optimisation. In addition, as was stated by Balcombe [35], animals are also capable of feeling positive emotions, and having access to a bath might also be considered as a pleasurable activity, easy to offer to these animals.

Gazing was observed 1.5% of the time, showing how these animals, even when kept in captive conditions, still have the instinct of protecting the group. Like the wild boar, they keep an eye on their surroundings and start a voice call or a run-away reaction if they hear or see anything considered as a threat. Gazing was considered the vigilant basic behaviour of pigs, but very little information regarding their anti-predator techniques has been described. Typically, it is assumed that a head-up, standing position covered this intention. Although animals may be able to detect predators even when not overtly vigilant, it has been shown that their ability is greater when they raise their heads [36]. In this study, when the position within the group was studied, Iberian pigs were more engaged in movement and vigilance activities when they were in the peripheral space of the groups, whereas the centrally located animals spent more time resting. Vigilance is a natural behaviour of prey species, in which they need to be alerted to detect predators or potential risks and alert the rest of the group. It is a costly strategy because it is traded off with other activities like feeding. Anti-predator behaviour and alarm responses have been studied in different species [31], like the bighorn sheep (*Ovis canadensis*) [37] or feral horses (*Equus ferus*) [38]. In meerkats (*Suricata suricatta*), individuals take successive guarding turns [39], but in chamois (*Rupicapra pyrenaica*), there are differences in the roles, depending on the status and the age of the animals [19]. In wild boars (*Sus scrofa*), synchronised vigilance is related to group size and risk factors [20], and in domestic pigs (*Sus scrofa domesticus*), adults employ different alarm barks than those of juveniles [40]. In the present study, all animals had the same age, but a difference in the activity was detected between animals in central positions versus those in the peripheral area. Further studies are needed to assess which factors determine the presence of animals in the periphery of the group and how stable this situation is across time.

In the present study, pigs dedicated more time of the day to exploratory activities than to social behaviours [41]. Accordingly, in the present study, and using the distinction between positive and negative social behaviour defined in Welfare Quality [27], pigs were seen 0.6% and 0.4% of the time showing social negative and positive behaviours, respectively. This was lower than what Temple et al. [7] (2.3% positive and 1% negative) or Stolba and Wood-Gush [42] (3% positive) found. Even if having more negative than positive social frequencies can be regarded as a problem, as Temple et al. [7] stated, positive social interactions might be reduced if enough space is provided to the animals (greater distances between them) or addressed at other hours of the day not observed in the present study (i.e., at night or from 13:30 h to 17:00 h, when very few observations were carried out). Temple et al. [7] found, in Iberian pigs reared in intensive conditions, six times more positive social behaviour than the same breed in extensive conditions, concluding that, probably, what is defined as social positive behaviour could sometimes be considered as not so positive. The Welfare Quality protocol, to avoid this bias, considers the presence of negative social behaviour out of the total social behaviour [27], assuming that in intensive conditions, the negative social behaviour is usually less than 50% of the total social contacts [7]. However, the results of the present study show that in extensive conditions, this assumption should be further studied. In fact, social behaviour could appear at specific moments when competition for a specific resource (i.e., area for resting) is important, and in these cases, a negative social behaviour would be expected instead of a positive contact. It is important to take into account that a higher incidence of negative social behaviour was found in spring, in comparison to the rest of the seasons, which can be attributed to the age of the animals, as it is in this season when piglets from different litters were mixed for first time into a single group. Another factor to consider is the weather. Cloudy days increased positive and negative social behaviour, in comparison to sunny days, rainy days increased only positive social behaviour and foggy days just negative social behaviour. In fact, the weather affected other behaviours. Active behaviours increased, especially when it rained, perhaps due to an increase in the humidity of the field. According to Hodgkinson et al. [32], loose and humid soils make pig motivation for rooting higher. Graves [43] also observed this influence of rainy and cloudy days in pigs, describing activity in pigs only during early in the morning and late at night under warm conditions. In contrast, during foggy days, pigs spent more than 82% of the time resting. This could be explained by the fact that they are prey animals and could feel more unsafe under these conditions. In addition, when partly cloudy, pigs spent 70% of the time resting, higher than the 56% found during sunny days or the 50% found during cloudy days. This could suggest that stable conditions, whether just sunny or just cloudy, would promote more activity than a changing situation, such as partly cloudy. In any case, in the present study, greater activity during rainy days was linked to more positive social behaviour, and reduced activity during foggy days was linked to an increase in negative social behaviour. Therefore, in areas with a high presence of fog throughout the year, with pigs in outdoor conditions, a lower level of activity and higher incidences of negative social behaviours could be expected in comparison with other areas. 

There were no significant differences between sexes or sexual status of the animals. There has been a long history of farmers assuring that female castration was a “must” in order to obtain good-quality products because entire females are more active during oestrus and consequently lose weight instead of putting it on [44,45]. In the present study, it was not measured whether a female was in oestrus or not, but, in general, no differences were found among any type of female (castrated surgically, vaccinated against GnRF or entire). This does not invalidate the fact that these differences could exist at specific moments, but in the present study, these differences were not found. In consequence, immunocastration could be a good alternative to entire females for farms with the presence of wild boar, where accidental mounting should be prevented. In fact, Martinez-Macipe et al. [22] found no differences in final-product quality among entire, castrated or vaccinated against GnRH female Iberian pigs reared in extensive conditions, and the present study indicates few differences among the different types of females in activity budgets.

## 5. Conclusions

During the observation periods of the present study, Iberian pigs spent 56% of the time resting and 28% exploring, as a mean, for the entire productive cycle. However, whether feeding them or not, the weather and the position within the group affected these activities. During the montanera period, Iberian pigs explored much more than during the rest of the year, but, in contrast to summer, they did not use water points for bathing. Animals staying in the centre of the group were more frequently resting than were the animals in the peripheral, which were more focused on movement and vigilance behaviours. Intact females did not present a different behaviour than did castrated or vaccinated against GnRH females.

## Figures and Tables

**Figure 1 animals-10-00798-f001:**
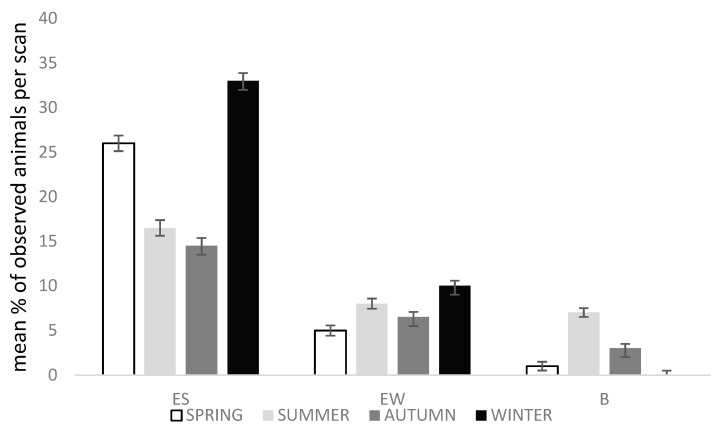
Mean % of observed animals per scan ± Standard Error of the Mean (SEM) of the explore standing (ES), explore walking (EW) and bathing (B) behaviours during the four seasons (spring, summer, autumn, winter) in relation to a total of 1439 observations. Analysed with general models of the Statistical Analysis System (SAS) (Poisson or negative binomial distribution).

**Figure 2 animals-10-00798-f002:**
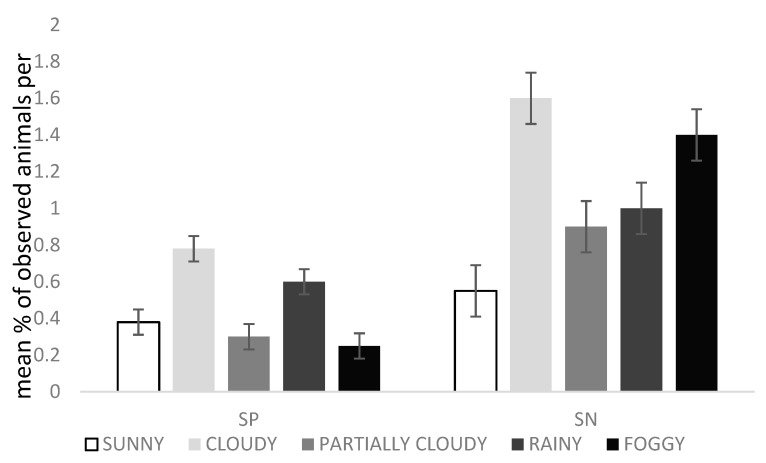
Mean % of observed animals per scan ± SEM of the social positive (SP) and social negative (SN) behaviours during the different weather conditions (sunny, cloudy, partly cloudy, rainy and foggy) in relation to a total of 1439 observations. Analysed with general models of SAS (Poisson or negative binomial distribution).

**Figure 3 animals-10-00798-f003:**
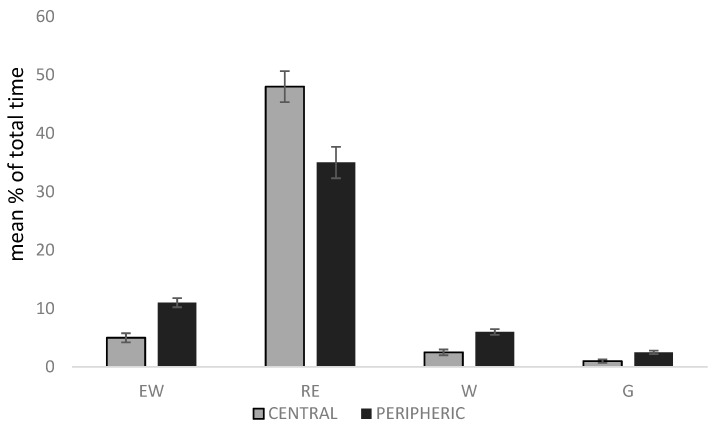
Mean ± SEM % of total time of the focal samplings that animals dedicated to resting (RE), explore walking (EW), walking (W) and gazing (G) according to their situation (central or peripheral) inside the group in relation to a total of 1247 observations. Analysed with general models of SAS (Poisson or negative binomial distribution).

**Table 1 animals-10-00798-t001:** Number of visits to each farm, percentage of purebred, quantity of pigs in the first and second year, available area during the year and during montanera, dates of montanera start for the first and second year of study.

Farm	Visits	% Purebred	Pigs at 1st year	Pigs at 2nd year	Area During the Year	Area during Montanera	Date 1st Montanera	Date 2nd Montanera
1	28	50%	94	40	7 Ha	12 Ha ^1^	13/11/2012	14/11/2013
2	3	50%	0	200	NA^2^	90 Ha	NA^2^	18/11/2013
3	44	100%	127	87	12 Ha	180 Ha	21/11/2012	19/11/2013
4	15	100%	145	0	8 Ha	30 Ha ^1^	23/11/2012	NA^2^
5	26	100%	140	200	3 Ha	60 Ha	14/12/2012	06/11/2013
6	17	50%	118	0	10 Ha	60 Ha	12/11/2012	NA^2^

^1^ Not all of the initial animals ended up in the montanera area studied but rather in another one. ^2^ NA means not assessed.

**Table 2 animals-10-00798-t002:** Behaviour description for scan and focal sampling.

Behaviour	Description ^1^
Exploring walking (EW)	Walks with the head low and nose at ground level. Includes feeding behaviour.
Exploring standing (ES)	Stands with the head low and nose at ground level. Includes feeding behaviour.
Resting (RE)	Lying down, laterally or ventrally.
^1^ Social positive (SP)	Sniffing, nosing, licking and moving gently away from the animal without aggressive or flight reaction from this individual.
^1^ Social negative (SN)	Aggressive behaviour, including biting, or aggressive social behaviour with a response from the disturbed animal.
Walk (W)	Moves with head up.
Run (RU)	Moves with head up and running.
Bath (B)	Enters the basin walking, with only the feet inside or more parts of the body, or swims.
Gazing (G)	Standing, with the head up and glance/look fixed.
Drink (D)	Takes water with the tongue and swallows it.
Other behaviours (O)	Urinates, defecates, coughs or others not listed.

^1^ Defined according to the Welfare Quality assessment protocol for pigs [27].

**Table 3 animals-10-00798-t003:** Temperature-humidity index (THI) and mean wind speed for each season from 2012 until 2014.

Season	Mean THI	Minimum THI	Maximum THI	Mean Wind Speed (km/h)
Spring 2012	24.03	21.50	25.83	2.2
Summer 2012	24.19	17.57	32.87	1.5
Autumn 2012	17.70	9.59	28.49	1.9
Winter 2012–2013	11.21	6.48	15.01	4
Spring 2013	15.20	8.82	25.05	1.8
Summer 2013	23.31	16.54	29.23	1.7
Autumn 2013	17.60	12.12	23.49	2.6
Winter 2013–2014	13.63	8.63	17.78	10.8

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
