# Peer review of "Environmental and Management Factors Affecting the Time Budgets of Free-Ranging Iberian Pigs Reared in Spain"

_animals, 2020, doi:10.3390/ani10050798_

Round 1

Reviewer 1 Report

Well done on greatly improving this paper.  You have worked hard.  Pig welfare and behaviour is a very important topic and it was fascinating to read your paper.  Good work improving the methodology and limitations.

Author Response

Thanks so much.

Reviewer 2 Report

Summary

The aim of this paper it is evaluating factors influencing the activity of Iberian pigs reared in outdoor conditions during different periods of the year, they were studied in six farms for 2 years.

Their main contributions describe as main activities resting and exploratory behaviour, which are influenced by the position of the animals in the group, weather and period of the year.

Broad comments

The study gives behavioural data about outdoor rearing in Iberian pigs in the southwest of Spain. The conclusions are supported by the data and give interesting results, relating behaviour and seasonal, climatic and spatial position within the group with the behavioural repertoire.

  1. I found the article well organized and clear in the presented manner. However, I would complete information for clarity in the statistical analysis. I suggest including the statistical unit used for the statistical analyses (L206). In the same way, the figures of the results should be self-explicative. So, I suggest including which are measures used for variance SD or SEM in the captions, the statistical test and the number of observations.

  1. In the discussion, I would appreciate a comment regarding the results of weather conditions and why could partially cloud weather have higher (L264) resting periods than sunny, but lower than cloudy and rainy days, independently of positive or negative social behaviours (L377).

Specific comments

  1. L271 Please include information in the captions in the same order than appears in the figure, from left to right. First social positive and after social negative, it is easier and faster to read.

  1. For clarity use the same terms in the figures and in the text, frequency appears in the figures as behavioural parameter, however it is not defined in this way in the material and methods. So please, be consistent with terms. (L255, L271).  

  1. Numbers of figures seem wrong and doesn’t match with the text, please correct number of figures “4” (L271) and “5” (L286)

  1. L292, between sexes or between males/females or females/males seems clearer than between the different sexes

  1. L100, welfare instead of welfar

  1. L212, focal sampling instead of scan, please be consistent with terms for behavioural observation, it is misleading if you say scan samplings taken individually and after focal sampling.

  1. I suggest to ad Lima 1998 as a key reference in the non-lethal effects of predator prey interactions (L346).

  1. Please, change yellow highlighted text to normal text, it doesn’t seem appropriate for reviewing

Author Response

Reviewer 2: I would complete information for clarity in the statistical analysis. I suggest including the statistical unit used for the statistical analyses (L206). In the same way, the figures of the results should be self-explicative. So, I suggest including which are measures used for variance SD or SEM in the captions, the statistical test and the number of observations.

Authors: In relation to the units, this information is now added: “In the case of scan sampling the unit was the group of animals observed and in the case of focal sampling the unit was the individual observed”. In relation to the captions of the figures, the information that SEM was used, the number of observations, and the statistical analysis (i.e. Analysed with General models of SAS (Poisson or negative binomial distribution) were added.

Reviewer 2: In the discussion, I would appreciate a comment regarding the results of weather conditions and why could partially cloud weather have higher (L264) resting periods than sunny, but lower than cloudy and rainy days, independently of positive or negative social behaviours (L377).

Authors: Some discussion was added in relation to the effect of partially cloud weather in comparison to sunny or cloudy weather: “In addition, when partly cloudy, pigs spent 70% of the time resting, higher than the 56% found during sunny days or the 50% of cloudy days. This could suggest that stable conditions, just sunny or just cloudy, would promote more activity than a changing situation, such as partly cloudy”

Reviewer 2: L271 Please include information in the captions in the same order than appears in the figure, from left to right. First social positive and after social negative, it is easier and faster to read.

Authors: Changed.

Reviewer 2: For clarity use the same terms in the figures and in the text, frequency appears in the figures as behavioural parameter, however it is not defined in this way in the material and methods. So please, be consistent with terms. (L255, L271).  

Authors: The word frequency was deleted from these two captions to avoid misunderstandings.

Reviewer 2: Numbers of figures seem wrong and doesn’t match with the text, please correct number of figures “4” (L271) and “5” (L286)

Authors: Changed.

Reviewer 2: L292, between sexes or between males/females or females/males seems clearer than between the different sexes

Authors: Between the different sexes was changed to between males and females.

Reviewer 2: L100, welfare instead of welfar

Authors: Corrected.

Reviewer 2: L212, focal sampling instead of scan, please be consistent with terms for behavioural observation, it is misleading if you say scan samplings taken individually and after focal sampling.

Authors: Sorry, this was a mistake. We were talking all time of focal sampling in this case. This is now corrected.

Reviewer 2: I suggest to ad Lima 1998 as a key reference in the non-lethal effects of predator prey interactions (L346).

Authors: Added.

Reviewer 2: Please, change yellow highlighted text to normal text, it doesn’t seem appropriate for reviewing

Authors: Changed.  

Reviewer 2: I would complete information for clarity in the statistical analysis. I suggest including the statistical unit used for the statistical analyses (L206). In the same way, the figures of the results should be self-explicative. So, I suggest including which are measures used for variance SD or SEM in the captions, the statistical test and the number of observations.

Authors: In relation to the units, this information is now added: “In the case of scan sampling the unit was the group of animals observed and in the case of focal sampling the unit was the individual observed”. In relation to the captions of the figures, the information that SEM was used, the number of observations, and the statistical analysis (i.e. Analysed with General models of SAS (Poisson or negative binomial distribution) were added.

Reviewer 2: In the discussion, I would appreciate a comment regarding the results of weather conditions and why could partially cloud weather have higher (L264) resting periods than sunny, but lower than cloudy and rainy days, independently of positive or negative social behaviours (L377).

Authors: Some discussion was added in relation to the effect of partially cloud weather in comparison to sunny or cloudy weather: “In addition, when partly cloudy, pigs spent 70% of the time resting, higher than the 56% found during sunny days or the 50% of cloudy days. This could suggest that stable conditions, just sunny or just cloudy, would promote more activity than a changing situation, such as partly cloudy”

Reviewer 2: L271 Please include information in the captions in the same order than appears in the figure, from left to right. First social positive and after social negative, it is easier and faster to read.

Authors: Changed.

Reviewer 2: For clarity use the same terms in the figures and in the text, frequency appears in the figures as behavioural parameter, however it is not defined in this way in the material and methods. So please, be consistent with terms. (L255, L271).  

Authors: The word frequency was deleted from these two captions to avoid misunderstandings.

Reviewer 2: Numbers of figures seem wrong and doesn’t match with the text, please correct number of figures “4” (L271) and “5” (L286)

Authors: Changed.

Reviewer 2: L292, between sexes or between males/females or females/males seems clearer than between the different sexes

Authors: Between the different sexes was changed to between males and females.

Reviewer 2: L100, welfare instead of welfar

Authors: Corrected.

Reviewer 2: L212, focal sampling instead of scan, please be consistent with terms for behavioural observation, it is misleading if you say scan samplings taken individually and after focal sampling.

Authors: Sorry, this was a mistake. We were talking all time of focal sampling in this case. This is now corrected.

Reviewer 2: I suggest to ad Lima 1998 as a key reference in the non-lethal effects of predator prey interactions (L346).

Authors: Added.

Reviewer 2: Please, change yellow highlighted text to normal text, it doesn’t seem appropriate for reviewing

Authors: Changed.  

Reviewer 3 Report

Comments Animals 726291

In this manuscript, the authors investigated the factors influencing the activity of Iberian pigs, reared free-range. They assessed the activity behaviours in six different farms, with management and climatic variations. The paper is clearly motivated and written, and the insight reported is a useful contribution for the evaluation of welfare in Iberian pigs, using a non-invasive method. However, the generalization of the results (and therefore the utility) to other breeds of pigs is difficult, since the type and environment of breeding of the Iberian pigs is peculiar and limited to a specific area of Spain. Please see below for a few comments that the authors could consider for revisions:

Simple summary

Lines 11-12: This sentence in not clear and introduce the concept of ‘normal behaviour’. What the authors mean by ‘normal behaviour’? A behaviour included in the ethogram of the species? Could please the authors clarify this point?

Abstract

I suggest removing the p-values in the abstract.

Introduction

Lines 57-59: I suggest adding references to this sentence.

Line 61: could the authors add a definition of ‘typical behaviour’?

Lines 65-66: could please the authors clarify the link between the welfare and the productive efficiency?

Lines 69-71: I suggest adding references to this sentence

Line 77: Overall, the authors wrote about extensive and semi-extensive systems. Could they please specify the difference between the two systems?

Lines 83-84: I suggest adding more examples on this topic, in addition to the chamois. Are there examples of pigs (not only Iberian breed) or phylogenetically similar species (for example wild boars)?

Line 86: Could the authors specify what kind of system?

Lines 101-103: This sentence is superfluous; I suggest removing it.

Material and Methods

Lines 128-137: I suggest inserting all this information into a table.

Lines 157-162: Did the influence of the presence of the observer the same at the beginning of the study? For example, for the first observations in all the farms (i.e., not only the first 10-15 minutes for each visit). Could the authors clarify this point?

Lines 163-164: This sentence needs a reference (e.g., 'Observational Study of Behavior: Sampling Methods' by Altman)

Line 169 and line 182: The authors indicate 1439 scan samples and 1247 focal sampling. How many for each farm?

Line 187: hereafter, I suggest replacing the term ‘atmospheric’ with ‘weather’,

Line 189: hereafter, I suggest replacing the term ‘factor’ with ‘index’

Lines 189-195: I think that THI is the more useful index to assess the heat stress, and consequently the animal’s welfare. However, the authors computed the index, and they don’t use it to evaluate behavioural activity of pigs. I think it's a missed opportunity, and I suggest (if it is possible) to compute the relationship between the index and the activity behaviours of pigs.

Line 212: Please replace ‘scan’ with ‘focal’.

Results

I am a little concerned about the treatment of farm 2 (low number of visits, missing data on areas, etc...) Could the authors please test and state the sensitivity of your findings when farm 2 is excluded from the analysis?

Overall, I suggest assessing if the social behaviours (positive and negative) are influenced by the density of animals (numbers of animals/available space).

Discussion

Lines 294, 316, 356: The sentence ‘during the observation of the present study’ is repeated too many times.

Lines 295, 299 etc… Please, add ‘et al.’ when a work with multiple authors is cited

Lines 317-320: I think that an important point could be to see how the use of the bath area is influenced by the THI.

Author Response

Reviewer 3: Lines 11-12: This sentence in not clear and introduce the concept of ‘normal behaviour’. What the authors mean by ‘normal behaviour’? A behaviour included in the ethogram of the species? Could please the authors clarify this point?

Authors: This sentence has been changed to: “To perform a specific behaviour could be critical when animal welfare is considered…”

Reviewer 3: I suggest removing the p-values in the abstract.

Authors: Removed.

Reviewer 3: Lines 57-59: I suggest adding references to this sentence.

Authors: Two references were added:

“Mellor, D.J. Updating Animal Welfare thinking: Moving beyond the “Five freedoms” towards “a life worth living”. Animals 2016, 6, 21. https://doi.org/10.3390/ani.6030021;

“Blokhuis, H.J., Veissier, I., Miere, M., Jones, R.B. The Welfare Quality project and beyond: Safeguarding animal well-being. Acta Agric. Scand. Section A, Anim. Sci. 2010, 60, 129-140.”

Reviewer 3: Line 61: could the authors add a definition of ‘typical behaviour’?

Authors: The sentence has been changed to: “As behaviour is regarded as a welfare indicator, it is important to know how Iberian pigs behave in their traditional environment to be able to detect welfare problems in the future”.

Reviewer 3: Lines 65-66: could please the authors clarify the link between the welfare and the productive efficiency?

Authors: A new reference was added for this purpose:

Dalmau, A., Duarte Borges, T., de Mercado, E., Gonzalez, I., Mateos-San Juan, A., Huerta-Jiménez, M., Gómez-Izquierdo, E., Lizardo, R., Pallisera, J., Borrisser-Pairó, F., Esteve-García, E., Panella-Riera, N., Ovejero, I. Effect of environmental temperature, floor type and breed on skatole and indole concentrations in fat of females, immune-castrated and entire males. Livestock Science. 2019. 220: 46-51”.

Reviewer 3: Lines 69-71: I suggest adding references to this sentence

Authors: We changed the sentence to: “Environmental and management factors, such as dietary supplementation or castration, may change the expression of some behaviour patterns of Iberian pigs reared outdoors, such as looking for food or mating behaviour. “

Reviewer 3: Line 77: Overall, the authors wrote about extensive and semi-extensive systems. Could they please specify the difference between the two systems?

Authors: In fact, the term semi-extensive is used in the paper only once and refered to the reference 15, where this term is used and we included it because we preferred to be precise on that: Jordana Rivero, M., Rodríguez-Estévez, V., Pietrosemoli, S., Carballo, C., Cooke, A.S., Grete Kongsted, A. Forage consumption and its effects on the performance of growing swine – Discussed in relation to European wild boar (Sus Scrofa L.) in semi-extensive systems: A review. Animals 2019, 9, 457-477. https://doi.org/10.3390/ani9070457

Reviewer 3: Lines 83-84: I suggest adding more examples on this topic, in addition to the chamois. Are there examples of pigs (not only Iberian breed) or phylogenetically similar species (for example wild boars)?

Authors: This sentence, and reference, has been added. However, in wild boars (Sus scrofa), synchronized vigilance is related to group size and risk factors [20]

Podgórski, T., De Jong, S., Bubnicki, J.W., Kuijper, D.P.J., Churski, M., JÈ©drzejewska, B. Drivers of synchronized vigilance in wild boar groups. Behav. Ecol. 2016, 27, 1097–1103. https://doi.org/10.1093/beheco/arw016

Reviewer 3: Line 86: Could the authors specify what kind of system?

Authors: The sentence has been changed to: “In the traditional production system for Iberian pig, both males and females are castrated”.

Reviewer 3: Lines 101-103: This sentence is superfluous; I suggest removing it.

Authors: Deleted.

Reviewer 3: Lines 128-137: I suggest inserting all this information into a table.

Authors: We did it in the past, but another reviewer ask us to simplify the table and put this information in text format. Comparing both systems, we really think that it is better in the way it is presented now.

Reviewer 3: Lines 157-162: Did the influence of the presence of the observer the same at the beginning of the study? For example, for the first observations in all the farms (i.e., not only the first 10-15 minutes for each visit). Could the authors clarify this point?

Authors: It is important to take into account that animals had thousands of square meters to their availability with trees, fences and hills and we tried to assess behaviour with binoculars. Although we can not discard that the last day of observation in a farm the animals were more used to our presence than the first one, there was other confounding factors for a clear observation of this fact. For instance, inside the same farm the distance to observe the animals could be different if they were in the pond or they were in the area with the closest forest or on top of a hill… So, we did not observe a clear effect on day of observation about the fact the animals felt disturbed, but probably, because the conditions of observation were not homogeneous from day to day as you need to adapt your observing area to the area where the animals are this specific day.

Reviewer 3: Lines 163-164: This sentence needs a reference (e.g., 'Observational Study of Behavior: Sampling Methods' by Altman)

Authors: A reference has been added:

Altmann, J. Observational study of behaviour: sampling methods. Behaviour, 1974, 49, 227-267.

Reviewer: Line 169 and line 182: The authors indicate 1439 scan samples and 1247 focal sampling. How many for each farm?

Authors: This information is now added for scans (310, 31, 468, 162, 290 and 177 for farms 1, 2, 3, 4, 5 and 6, respectively) and focals (286, 32, 393, 131, 261 and 144 for farms 1, 2, 3, 4, 5 and 6, respectively).

Reviewer 3: Line 187: hereafter, I suggest replacing the term ‘atmospheric’ with ‘weather’,

Authors: Replaced

Reviewer 3: Line 189: hereafter, I suggest replacing the term ‘factor’ with ‘index’

Authors: Replaced

Reviewer 3: Lines 189-195: I think that THI is the more useful index to assess the heat stress, and consequently the animal’s welfare. However, the authors computed the index, and they don’t use it to evaluate behavioural activity of pigs. I think it's a missed opportunity, and I suggest (if it is possible) to compute the relationship between the index and the activity behaviours of pigs.

Authors: We included the relationship between THI and use of the bathing area as suggested by the reviewer in lines 317-230. In relation to THI, when ranged from 6 to 12, animals were seen bathing only in one occasion (0.01%), being a 0.62% when ranging from 13 to 19, 4.41% when ranging from 20 to 26 and 12% when ranging from 27 to 33.

Reviewer 3: Line 212: Please replace ‘scan’ with ‘focal’.

Authors: Replaced.

Reviewer 3: I am a little concerned about the treatment of farm 2 (low number of visits, missing data on areas, etc...) Could the authors please test and state the sensitivity of your findings when farm 2 is excluded from the analysis?

Authors: In fact, we were as well concerned by that when we were working the data at the beginning, before writing the paper, but when we compared the different models with or without these data (the farm effect, with 5 or 6 is included in the model) we did not lost too much sensitivity and in contrast we had 31 extra scans and 32 extra focals in the analysis, so we decided to include it.

Reviewer 3: Overall, I suggest assessing if the social behaviours (positive and negative) are influenced by the density of animals (numbers of animals/available space).

Authors: In fact, we have two types of densities to take into account under these circumstances. The first one, is the total surface available for animals, that for instance, in the montanera season ranged from 1282 m2 per pig to 20000 m2 per pig, and this effect is in fact inside the farm factor. The second one have relation with the real use in a specific moment of the available space the animals have, because as gregarious species they are, they decide (in group) to use specific areas inside this vast territory they have, and, as commented previously, they can change from one day to another the area selected. So, the real density should be calculated to this self-restricted area where the animals decided to eat, rest or walk in a specific moment. Unfortunately, although this could be an interesting information (to see the size of the area used from one extreme to another where a group is, at different moments) we don’t have this information.

Reviewer 3: Lines 294, 316, 356: The sentence ‘during the observation of the present study’ is repeated too many times.

Authors: In the two last cases it was substituted for In the present study.

Reviewer 3: Lines 295, 299 etc… Please, add ‘et al.’ when a work with multiple authors is cited

Authors: Done

Reviewer 3: Lines 317-320: I think that an important point could be to see how the use of the bath area is influenced by the THI.

Authors: According to the results showed previously, the next sentence was added: “In fact, below a THI of 20, animals seen bathing did not arrive to a mean value of 1%, but above 27 the mean value was 12%.”

Round 2

Reviewer 3 Report

Accept in present form

Author Response

Thanks

This manuscript is a resubmission of an earlier submission. The following is a list of the peer review reports and author responses from that submission.

Round 1

Reviewer 1 Report

I have read through the revised manuscript and do not believe it is substantially improved from the previous version. The authors have made only specific changes that were suggested, but have not taken on board the previous reviewer comments (both reviewers) that the whole paper needs to be carefully and thoroughly edited for it to be of a standard for publishing. The flaws relate both to the language and also to the logical flow of the paper. The figures are also still messy, and some areas of the paper lack adequate referencing. I am sorry but I am unable to recommend this paper for publication as I believe it is not yet of a suitable standard. 

Reviewer 2 Report

You have produced a well written paper on pig welfare and behaviour.  There were some strong points in your approach eg after entering the pigs' enclosure you allowed time for the animals to settle prior to recording the observations.  Recording your observations verbally onto a portable tape recorder was an interesting approach.  It was fascinating to read about the montanera period and good how the authors discussed limitations in the observational technique used ie explanation that it could be difficult differentiating individual animals.  Some constructive comments for the authors include that I feel that regardless of supplementary feeding or not, the pigs may have avoided bathing in the montanera period because it coincides with cooler temperatures (Winter); also, the description of female pigs being 'castrated' (eg line 87 and elsewhere) is technically incorrect (although I understood what you meant).  For females, the term 'speyed' or 'spayed' is more correct, or even 'neutered'.  

Minor errors to be corrected include:  line 115 'outdoor' should be 'outdoors' and line 401 'o' should be 'or'.

Otherwise, well done.  I will recommend your paper is published.  I appreciate the hard work involved in such a study.  

The highlighting needs to be removed prior to resubmission.

Good luck and thanks.

Reviewer 3 Report

The paper is conceptually very interesting, as having an ethogram of Iberian pigs managed outdoors could provide interesting insights when compared with the behaviour of free ranging wild boars, or of domesticated pigs under more controlled conditions. Unfortunately the study objectives are not presented clearly, and the experimental design and execution of the study are poorly done.

An essential part of the scientific method is that the methods are clearly stated, such that the study could theoretically be repeated by other scientists. The methods described here are unclear and irregular. Pigs were studied at irregular time periods (sometimes the morning, sometimes the afternoon), but no indication of the frequency of either morning or afternoon observations is given. The scan and focal sampling procedures are not clearly presented. Gender differences are discussed, but no data on the number of entire males, castrates, immunocastrates, entire females or sterilized females in each group, or overall, is given.

The ethogram used is sourced from the Welfare Quality protocol. If the objective was to describe the behaviour of outdoor pigs, why was a welfare audit used as the basis for the ethogram? If the goal is to describe the behaviour of the animals, why were the pigs observed for only 2 hours per day, and usually in the morning? Presenting time budgets based on this narrow window of observation is misleading.

The data collected may be of some value, but cleaning of the data is needed (for example, to remove or better define irregular observation times). The objectives need to be revisited and clarified, followed by re-analysis and a clear presentation of the results.